# DLER: Doing Length pEnalty Right - Incentivizing More Intelligence per Token via Reinforcement Learning

## Abstract

Reasoning language models such as OpenAI-o1, DeepSeek-R1, and Qwen achieve strong performance via extended chains of thought but often generate unnecessarily long outputs. Maximizing intelligence per token—accuracy relative to response length—remains an open problem. We revisit reinforcement learning (RL) with the simplest length penalty—truncation—and show that accuracy degradation arises not from the lack of sophisticated penalties but from inadequate RL optimization. We identify three key challenges: (i) large bias in advantage estimation, (ii) entropy collapse, (iii) sparse reward signal. We address them with **D**oing **L**ength p**E**nalty **R**ight (**DLER**), a training recipe combining batch-wise reward normalization, higher clipping, dynamic sampling, and simple truncation length penalty. DLER achieves state-of-the-art accuracy–efficiency trade-offs, cutting output length by over 70% while surpassing all previous baseline accuracy. It also improves test-time scaling: compared to DeepSeek-R1-7B, DLER-7B generates multiple concise responses in parallel with 28% higher accuracy and lower latency. We further introduce Difficulty-Aware DLER, which adaptively tightens truncation on easier questions for additional efficiency gains. We also propose an update-selective merging method that preserves baseline accuracy while retaining the concise reasoning ability of the DLER model which is useful for scenarios where RL training data is scarce.

## 1 Introduction

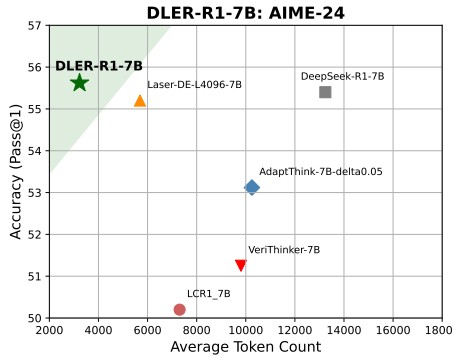

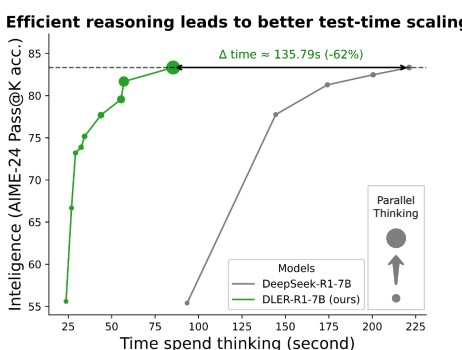

(a) DLER Training on DeepSeek-R1-7B

(b) DLER-R1-7B enable better test-time scaling

Figure 1: (a) DLER achieves state-of-the-art accuracy/length trade-offs, shortening CoT by up to 70% without losing accuracy. (b) On AIME-24, DLER-R1 models enable better test-time scaling. Results for 1.5B models are shown in Fig. 7 and Fig. 8 in the appendix.

Reasoning models such as OpenAI-o1 (Jaech et al., 2024), DeepSeek-R1 (Guo et al., 2025), and Qwen (Yang et al., 2025a) achieve strong performance through long chains of thought (CoT) (Yang et al., 2025b), but this comes at the cost of heavy token usage, higher latency, and redundant outputs for questions solvable with shorter responses. Therefore, how to maximize intelligence per token

remains an open research question. Recent work has addressed the inefficiency of extended reasoning by developing methods to reduce output length. These approaches fall into three categories: prompt engineering (Ma et al., 2025a), supervised fine-tuning (Lu et al., 2025; Chen et al., 2025; Liu et al., 2024; Xia et al., 2025; Ma et al., 2025b), and reinforcement learning (RL) (Fang et al., 2025; Liu et al., 2025b; Luo et al., 2025a; Hou et al., 2025; Aggarwal & Welleck, 2025). Among these, RL-based methods have emerged as the most principled approach for achieving optimal accuracy-efficiency trade-offs. These methods typically incorporate length penalties into the reward function to incentivize reasoning within predefined token budgets. However, despite demonstrating substantial reductions in reasoning length, existing approaches often suffer from accuracy degradation that varies significantly across tasks of different complexity levels—a limitation we hypothesize stems from suboptimal optimization techniques.

In this work, **we revisit reinforcement learning (RL) for reasoning efficiency by re-examining the simplest length penalty—truncation**, which assigns zero reward to responses exceeding a fixed limit. Prior RL methods using truncation often fail to recover accuracy; we find this stems not from the length penalty itself but from sub-optimal RL optimization techniques. Three issues drive this degradation: (1) biased advantage estimation under Group Relative Policy Optimization (GRPO) (Shao et al., 2024) due to substantial reward noise, especially in early state of training, as many responses are abruptly cut off and assigned zero reward, (2) persistent entropy collapse that hampers exploration of diverse reasoning paths, and (3) sparse reward signals arising from a large portion of prompts in each batch where all rollouts are truncated and thus assigned zero reward. We address these by adopting batch-wise normalization (Hu et al., 2025), higher clipping thresholds (Yu et al., 2025), to promote exploration via low-probability, high-entropy tokens, and curriculumized filtering to gradually introduce harder prompts Yu et al. (2025).

By combining all essential elements, our final training recipe, *Doing Length pEnalty **R**ight (DLER)*, achieves state-of-the-art accuracy-to-token efficiency. As illustrated in Fig. 1a, DLER fully recovers the accuracy drop while reducing the average response length by over 70%. This underscores key insights:

> **Key Insight 1:** It is not the sophisticated design of the length penalty that determines performance, but rather the choice of RL optimization algorithm. Even the simplest length truncation can achieve state-of-the-art accuracy-to-token efficiency when combined with our DLER recipe. See Sec. 4.2 for more details.

> **Key Insight 2:** We additionally apply DLER recipe to a variety of length penalties and find that they no longer push the frontier of accuracy-efficiency frontier, but instead serve as tools for fine-grained adjustment of the trade-offs. See Sec. 4.5 for more details.

We also benchmark test-time scaling by generating multiple responses in parallel. Fig. 1b shows that our DLER-R1-7B delivers significant improvements over the original DeepSeek-R1-7B, achieving a 27% accuracy gain (AIME-24) within the same wall-clock "thinking time." This marks an important shift in perspective: whereas recent efforts (Guo et al., 2025; Liu et al., 2025a; Yu et al., 2025) to enhance reasoning ability have largely pursued accuracy through increasingly long reasoning traces, our findings demonstrate that:

> **Key Insight 3:** Improving reasoning efficiency not only lowers the cost of single response but also enables superior test-time parallel scaling. See Sec. 4.4 for more details.

We further propose a difficulty-aware variant of DLER (**DA-DLER**), where the truncation target length is dynamically adjusted according to an estimate of the model's ability to solve the question. For questions that the model can already reliably answer within the target length, the target length is further shortened to encourage even shorter reasoning, while more challenging questions are allowed more tokens. DA-DLER can achieve an additional 15% and 11% reduction in response length on DeepSeek-R1-1.5B and 7B, respectively, further advancing the efficiency frontier.

We also provide a solution for practical scenarios where accessing the original RL training dataset is not feasible. Often, applying small-scale academic RL training datasets to proprietary models leads to accuracy degradation, and length penalties exacerbate this issue (Liu et al., 2025b) while remaining effective at reducing output length. To completely mitigate accuracy degradation without

access to the original dataset, we adopt an update-selective weight merging strategy that combines the original baseline model with the DLER-trained model. This method recovers all lost accuracy while still reducing output tokens by 47%.

> **Key Insight 4:** Weight merging allows for a better trade-off between accuracy and length reduction especially when accessing the original high-quality proprietary datasets is not an option. See Sec. 4.6 for more details.

## 2 PRELIMINARY

Reinforcement learning is widely applied to enhance the reasoning ability of modern LMs Comanici et al. (2025); Achiam et al. (2023), with GRPO (Shao et al., 2024) becoming popular for its efficiency in removing the critic model and using group-relative advantage estimation. This approach maintains token-level advantage estimation accuracy while significantly reducing the overhead. Specifically, for each question-answer pair $(q, a)$, the behavior policy $\pi_{\theta_{\text{old}}}$ samples a group of $G$ responses $\{o_i\}_{i=1}^{G}$. The advantage for the $i$-th response is then computed by normalizing the group-level rewards $\{R_i\}_{i=1}^{G}$ as:

$$A_{i,t} = \frac{R_i - \text{mean}(\{R_i\}_{i=1}^{G})}{\text{std}(\{R_i\}_{i=1}^{G})} \tag{1}$$

and the optimization objective is formulated as:

$$\mathcal{J}_{\text{GRPO}}(\theta) = \mathbb{E}_{(q,a)\sim D, \{o_i\}_{i=1}^{G}\sim\pi_{\theta_{\text{old}}}(\cdot|q)} \left[ \frac{1}{G}\sum_{i=1}^{G} \frac{1}{|o_i|}\sum_{t=1}^{|o_i|} \min\left(s_{i,t}(\theta)\,A_{i,t},\ \text{clip}(s_{i,t}(\theta), 1-\epsilon, 1+\epsilon)\,A_{i,t}\right)\right] \tag{2}$$

where $s_t(\theta) = \frac{\pi_\theta(o_t\,|\,q,o_{<t})}{\pi_{\theta_{\text{old}}}(o_t\,|\,q,o_{<t})}$ and $\epsilon$ is the clipping threshold. We omit the KL Loss for simplicity. Recent studies (Liu et al., 2025b; Fang et al., 2025; Hou et al., 2025; Aggarwal & Welleck, 2025), aiming to enhance reasoning efficiency, commonly adopt GRPO as the optimization algorithm, coupled with custom-designed length penalty rewards. Under this setup, the new reward $R'_i$ is generally formulated as: $R'_i = R_i + L_i$ where $R_i$ denotes the correctness reward, computed using rule-based heuristics, and $L_i$ represents the length penalty. Depending on the specific length penalty, $L_i$ may either impose a larger penalty on longer outputs or, in the case of the simple truncation penalty, assign a reward of zero to any response that exceeds a predefined length limit.

## 3 RE-EXAMINING THE SIMPLEST LENGTH PENALTY - TRUNCATION

Prior studies that aim to enhance training efficiency tend to treat the underlying policy optimization algorithm as a fixed, reliable component, often attributing improvements in accuracy-to-length ratio primarily to the design of length penalties. However, this overlooks the possibility that the optimization algorithm itself may introduce performance bottlenecks. In this work, we re-examine the structure of the policy optimization objective by adopting the simplest possible length penalty—truncation—which is simple enough to alleviate reward hacking and enables a focused analysis of how the optimization algorithm alone affects accuracy degradation.

### 3.1 MORE AGGRESSIVE TRUNCATION LEADS TO HIGHER GROUP-WISE REWARD VARIANCE

First, we find that truncation significantly increases reward variance. Using DeepSeek-R1-7B at step 0 with 16 rollouts per question and 512 prompts from the DeepScaleR-Preview-Dataset (Luo et al., 2025b), we measure advantage variance under truncation lengths of $\{4000, 8000, 12000, 16000\}$, yielding values $\{0.40, 0.32, 0.30, 0.29\}$. More aggressive truncation thus amplifies per-prompt variance and bias in advantage estimation (Eq. 1), leading to training instability (see Appendix B).

To address the increased bias introduced by truncation, we propose replacing GRPO's prompt-wise advantage normalization with global batch-wise advantage normalization, which is also used in (Hu et al., 2025). The idea is that by normalizing the advantages across the entire batch, we can mitigate the impact of outliers and ensure a more stable estimation of advantage variance. The advantage calculation for the $i$-th response after adopting batch-wise reward normalization becomes:

$$A_{i,t}^{\text{norm}} = \frac{A_{i,t} - \text{mean}_{\text{batch}}(A_{i,t})}{\text{std}_{\text{batch}}(A_{i,t})} \tag{3}$$

where $A_{i,t} = R'_i - \text{mean}(\{R'_i\}_{i=1}^{G})$ and we can see that the normalization is now done on batch level rather than local group level.

We compare GRPO and batch-wise reward normalization by training DeepSeek-R1-7B on the DeepScaleR-Preview-Dataset and evaluate every 10 steps on AIME-24; see Sec. 4.1 for full experimental details. We observe that while both methods reduce output length during training, GRPO exhibits a steady decline in accuracy, whereas batch-wise normalization starts to recover after about 100 steps, yielding roughly a 3% improvement under comparable token budgets. This highlights that GRPO's instability under truncation degrades performance, whereas batch-wise normalization stabilizes training and restores accuracy. See Fig. 9 in the appendix for the full training dynamics of accuracy and average response length of group-wise normalization and batch-wise normalization on the AIME-24 test set.

> **Key Finding 1:** Length truncation increases reward variance, leading to biased advantage estimates and degraded performance when using GRPO. Switching to batch-wise reward variance estimation mitigates this issue and improves performance.

## 3.2 Entropy Collapse Limits Exploration of Reasoning Paths

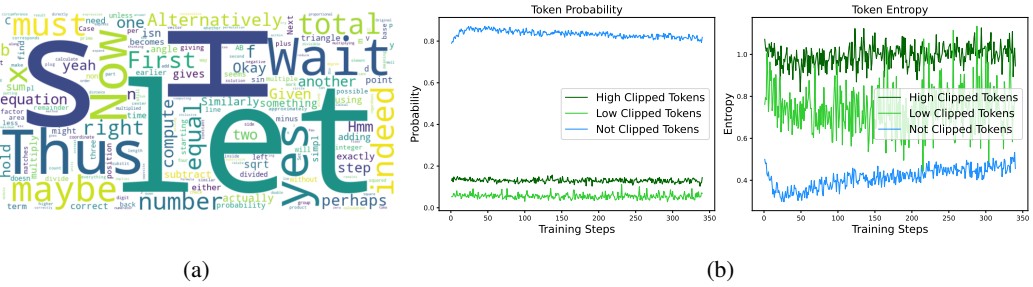

(a)                                                                (b)

Figure 2: **(a)** Word clouds of frequent tokens clipped by the high-threshold $(1 + \epsilon)$. **(b)** Clipped tokens have much lower probabilities and higher entropy simultaneously than unclipped ones.

Although switching from group-wise to batch-wise reward normalization improves accuracy, it does not fully restore performance. The model still suffers from entropy collapse (Yu et al., 2025; Liu et al., 2025a), where the model's output distribution becomes overly concentrated, causing the policy to prematurely focus on a narrow set of responses. This limits exploration, introduces bias in policy updates, and ultimately stalls training progress.

We suspect that this issue may be caused by the clipping on the importance sampling ratio $(s_t(\theta))$ in Eq. 2 which zeroes out gradients of clipped tokens. To investigate their impact, we analyzed the clipped tokens and found that although they account for only about 1% of all tokens but, as shown in Fig. 2a, are dominated by transitional cues like "Wait," "Hmm," "Alternatively," "thus," and "also," which often play an important role in determining response length and are crucial for structuring reasoning paths (Wang et al., 2025). More interestingly, as shown in Fig. 2b, we find there is a big overlap between low-probability tokens, high-entropy tokens, and clipped tokens. We find that clipped tokens—especially those clipped by the upper threshold $(1 + \epsilon)$—tend to have lower probabilities and higher entropy simultaneously. This finding also connects studies on low-probability tokens (Yang et al., 2025c) and high-entropy tokens (Wang et al., 2025) and provides a unified perspective to understand previous different findings on token importance for optimization.

By decoupling the lower and upper thresholds—previously set to the same value—and assigning a larger value to the upper threshold, the gradient update on those high entropy exploratory tokens are retained, allowing their gradients to propagate and fostering more diverse reasoning behaviors during training. We compare DeepSeek-R1-7B with batch-wise reward normalization, with and without the higher clipping strategy, using a truncation target length of 4000. As shown in Fig. 4, enabling a higher clipping threshold effectively mitigates entropy collapse: entropy not only avoids vanishing but increases after an initial drop, in contrast to the decline observed in DAPO (Yu et al., 2025) and ProRL (Liu et al., 2025a) without truncation penalties.

> **Key Finding 2:** The clipped tokens are often low-probability, high-entropy tokens that play a crucial role in exploration of reasoning paths and length control. Adopting a higher clipping threshold helps retain these tokens in gradient updates, thereby mitigating entropy collapse.

## 3.3 LENGTH PENALTY OVER-SPARSIFY TRAINING SIGNAL

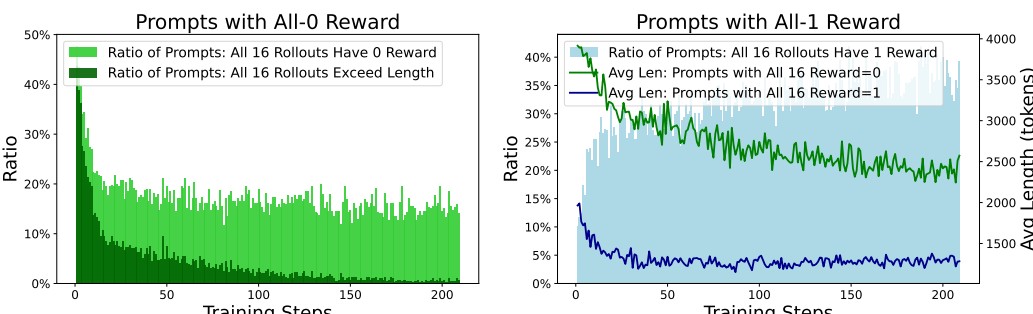

Figure 3: **Left**: Ratio of training prompts with all 16 rollouts receiving zero reward, including those caused by exceeding the truncation length. Around half of the prompts fall into this category early in training, weakening the signal and biasing the model toward easier prompts that model already know how to solve within the target length. **Right**: Ratio of training prompts with all 16 rollouts receiving reward score of one steadily increases, while average response length declines and remains markedly shorter than that for prompts whose rollouts all receive a reward of zero.

Another phenomenon we identify with the application of length penalty is the prevalence of zero-reward signals across training rollouts. Specifically, a substantial fraction of the prompts receive zero reward for all 16 rollouts, primarily because all 16 responses exceed the target length.

As shown in Fig. 3, nearly half of all prompts are affected at the start of training, where sparse, noisy feedback biases the model toward easier prompts it can already solve, reducing exploration and limiting accuracy gains. Later, prompts with all rollouts receiving positive reward grow to 40% of the batch. These prompts are typically easier, yielding consistently short responses, but their dominance causes the model to overfit to overly short outputs and underuse the target budget. This explains why a suboptimal RL run (Fig. 4(b)) plateaus at 2k tokens despite the maximum being 4k—the model has prematurely overfit to easy prompts. To mitigate this issue, we discard prompts whose rollouts all yield zero reward or all yield positive reward and resample until the target batch size is reached (Yu et al., 2025). This dynamic sampling strategy implicitly induces a curriculum, as it progressively incorporates harder examples that initially demand longer reasoning chains to solve. As evident from the training dynamics shown in Fig. 4(c), the model autonomously learns to rapidly reduce token usage in the early training stages, then gradually increases length to fully exploit the target token budget. This behavior is driven purely by the simple truncation length penalty. Without dynamic sampling, the model tends to plateau at a shorter length, as shown in Fig. 4(b), indicating that it has prematurely overfitted into a suboptimal local minimum.

> **Key Finding 3:** Dynamic sampling filters out prompts either too easy or too hard and responses either too long or too short, either of which can over-dominate the training batch and skew the reward shape. This leads to a more balanced training signal and enables the model to better utilize the target length budget.

## 3.4 COMBINING ALL INGREDIENTS: DO LENGTH PENALTY RIGHT

Building on our findings, we unify batch-wise reward normalization, a higher policy update clipping threshold, dynamic sampling to remove instances lacking balanced training signals, and a simple length truncation penalty into a comprehensive training recipe, which we term **DLER** (**D**oing **L**ength p**E**nalty **R**ight). We apply DLER to train DeepSeek-R1-7B for 450 steps and compare its training dynamics (Fig. 4) along with the trajectory of accuracy and response length on AIME-24 (Fig. 10). All ingredients work complementary to each other. Together, these components sys-

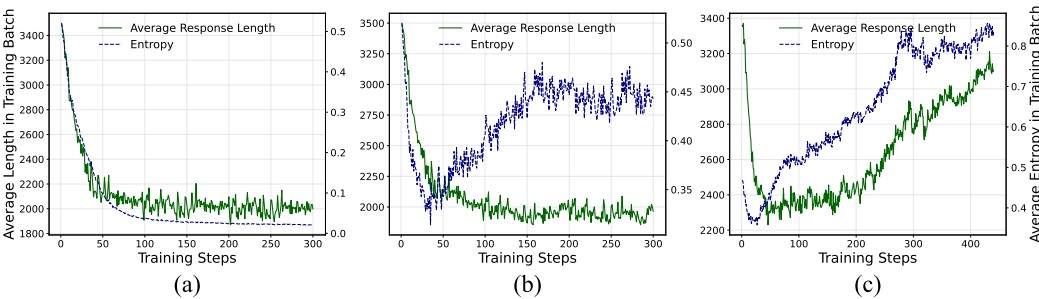

Figure 4: Average token entropy and response length per training batch across three RL runs: **(a)** batch-wise normalization, **(b)** batch-wise normalization with a higher clipping threshold, and **(c)** our DLER method. DLER not only resolves entropy collapse but also shows rising entropy and gradually increasing response length after an initial drop, suggesting active exploration under the length penalty, unlike the plateaued behavior of the baselines.

tematically address the optimization challenges identified in previous sections and synergistically contribute to DLER's superior performance.

### 3.5 DIFFICULTY-AWARE DLER

We also present a **D**ifficulty-**A**ware extension of **DLER** (**DA-DLER**) that improves efficiency by adaptively assigning truncation lengths according to question difficulty, yielding greater redundancy reduction than a fixed truncation scheme. Question difficulty is estimated from the correctness ratio of model responses, and truncation targets are dynamically adjusted across difficulty tiers. Concretely, for a question $q$ with a sampled response set $G$, the correctness ratio is defined as the fraction of correct responses in $G$. Each question is then categorized into one of $n$ difficulty levels $\{d_i\}_{i=1}^n$, each associated with a truncation length $\{l_i\}_{i=1}^{n+1}$. For instance, if the correctness ratio lies between $d_i$ and $d_{i+1}$, the truncation length $l_i$ is applied. DA-DLER pushes the boundary of reasoning efficiency by encouraging the model to solve questions it already handles reliably with even fewer reasoning steps, reducing unnecessary token usage.

## 4 EXPERIMENT

### 4.1 SETUP

We conduct experiments on DeepSeek-R1-1.5B/7B (Guo et al., 2025), widely used as baseline models by prior works (Chen et al., 2025; Liu et al., 2025b; Zhang et al., 2025; Aggarwal & Welleck, 2025; Luo et al., 2025a). Training uses the DeepScaleR-Preview-Dataset (Luo et al., 2025b) (40K competition math problems) with veRL (Sheng et al., 2024), following the original DeepSeek-R1 prompt format. We adopt 16 rollouts per prompt, batch size 512, and a truncation penalty target length of 4000 tokens; full hyperparameters are listed in Appendix D. The trained models are referred to as **DLER-R1-1.5B/7B**. We further continue training for 150 steps with the difficulty-aware recipe (DA-DLER), yielding **DA-DLER-R1-1.5B/7B**, where the difficulty threshold is set to a correctness ratio of 0.5 and truncation penalties of 2000 and 4000 tokens.

We compare against prior publicly released models trained on DeepSeek-R1, including: (1) Laser (Liu et al., 2025b), with difficulty-aware penalties (**Laser-DE-L4096-1.5B/7B**); (2) Adapt-Think (Zhang et al., 2025), which skips reasoning for easy questions (**AdaptThink-1.5B/7B**); (3) LC-R1 (Cheng et al., 2025), which adds a conciseness and compress reward (**LCR1-1.5B/7B**, trained on a different dataset); and (4) VeriThinker (Chen et al., 2025), an SFT method with improved self-reflection (**VeriThinker-7B**).

We evaluate all models on AIME-24 (MAA, 2024a), AMC 2022/2023 (MAA, 2024b), MATH (Hendrycks et al., 2021), Minerva (Lewkowycz et al., 2022), and Olympiad Bench (He et al., 2024), using vLLM with temperature 0.6, $top_p$=0.95, and max length 32k. For each prompt, we generate 16 samples and report average pass@1 accuracy.

## 4.2 MAIN RESULTS

Table 1: Comparison of DLER models and baseline models in terms of Pass@1 accuracy and corresponding average output length (tokens) across benchmarks.

| | MATH ↑ | Length ↓ | AIME-24 ↑ | Length ↓ | AMC ↑ | Length ↓ | Minerva ↑ | Length ↓ | Olympiad ↑ | Length ↓ | Total Avg ↓ |
|---|---|---|---|---|---|---|---|---|---|---|---|
| DeepSeek-R1-1.5B | 84.31 | 5500 | 29.79 | 16916 | 61.97 | 10967 | 38.41 | 7494 | 44.07 | 11620 | 10499 |
| LCR1-1.5B | 81.80 | 2612 | 21.04 | 9335 | 59.64 | 5377 | 40.64 | 2702 | 41.58 | 5876 | 5180 |
| Laser-DE-L4096-1.5B | 85.27 | 2685 | 30.62 | 8194 | 68.14 | 4890 | 42.69 | 3322 | 46.21 | 5323 | 4882 |
| AdaptThink-1.5B-delta0.05 | 82.26 | 1651 | 30.21 | 7550 | 61.37 | 3622 | 41.52 | **1745** | 42.50 | 4279 | 3769 |
| **DLER-R1-1.5B** | **86.95** | 1652 | **34.38** | 3551 | 70.48 | 2537 | 43.59 | 2029 | 48.31 | 2563 | **2466 (-77%)** |
| **DA-DLER-R1-1.5B** | 86.70 | **1484** | 34.37 | **2888** | **72.36** | **2154** | **44.89** | 1895 | **48.70** | 2109 | **2106 (-80%)** |
| DeepSeek-R1-7B | 93.60 | 3999 | 55.40 | 13241 | 82.90 | 7461 | 49.79 | 5199 | 58.21 | 8837 | 7747 |
| R1-VeriThinker-7B | 93.63 | 2591 | 51.25 | 9805 | 81.77 | 5611 | 46.14 | 2934 | 57.92 | 6470 | 5482 |
| LCR1-7B | 90.65 | 1534 | 50.20 | 7305 | 79.29 | 3609 | 50.32 | **1559** | 55.96 | 4352 | 3671 |
| Laser-DE-L4096-7B | 93.48 | 1759 | 55.20 | 5691 | 82.83 | 3262 | 50.22 | 1884 | 57.90 | 3451 | 3209 |
| AdaptThink-7B-delta0.05 | 91.38 | 2005 | 53.12 | 10250 | 81.47 | 5461 | 50.67 | 2522 | 56.96 | 6434 | 5334 |
| **DLER-R1-7B** | **94.21** | 1634 | **55.62** | 3230 | 84.41 | 2512 | **53.88** | 2058 | 60.48 | 2592 | **2405 (-69%)** |
| **DA-DLER-R1-7B** | 94.17 | **1481** | 53.90 | **2878** | **84.56** | **2286** | 53.60 | 1896 | **61.16** | 2296 | **2167 (-73%)** |

Table 1 compares DLER and DA-DLER with prior state-of-the-art baselines on five benchmarks—MATH, AIME-24, AMC, Minerva, and Olympiad—along with average response length. On DeepSeek-R1-1.5B, DLER-R1-1.5B achieves the highest accuracy on all benchmarks while reducing average length to 2466 tokens, over 4× shorter than the base model and up to 51% shorter than previous methods. For example, it scores 86.95 on MATH, 34.38 on AIME, and 48.31 on Olympiad, outperforming Laser-DE by 1.68 ∼ 3.76 points. DA-DLER-R1-1.5B further improves efficiency, cutting length by an additional 15% (2466 to 2106) while maintaining accuracy.

Similar trends hold for the 7B model. DLER-R1-7B reaches 94.21 on MATH, 55.62 on AIME, and 84.41 on AMC, with average length 2405 tokens—69% shorter than the base model and 25% shorter than Laser-DE—while also improving accuracy. DA-DLER-R1-7B reduces length by a further 12% without loss of accuracy. Overall, DLER-models provides the best accuracy–efficiency trade-off across both model sizes, unlike prior methods that either sacrifice accuracy or retain long responses. Consistent gains on challenging datasets such as AIME-24 and Olympiad highlight the robustness of our recipe.

We further analyze DLER's impact on model diversity and exploration through entropy analysis (Appendix G.1), and evaluate its role in mitigating overthinking via reasoning trace analysis (Appendix G.2).

## 4.3 PERFORMANCE UNDER DIFFERENT TEST-TIME SCALING SETTINGS

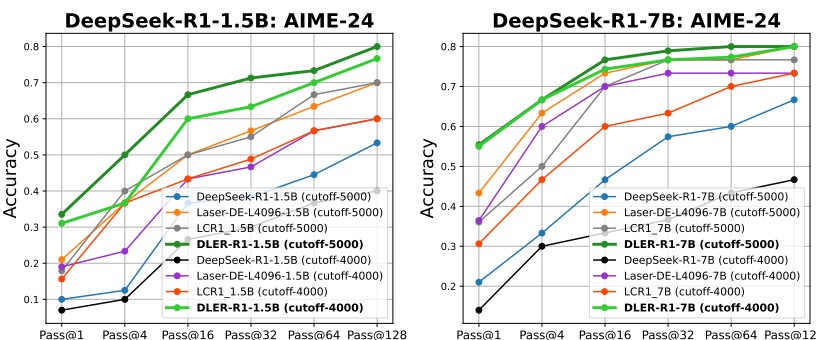

Figure 5: Pass@K accuracy of DLER-1.5B/7B versus baselines on AIME-24 under different length cutoffs. DLER consistently outperforms all baselines across different test-time budget settings.

We compare DLER-R1-1.5B/7B with Laser-DE, LCR1, and the original DeepSeek-R1 under different test-time scaling by evaluating Pass@K across response length cutoffs. As shown in Fig. 5, on AIME-24 with cutoffs of 4000 and 5000 tokens, DLER consistently outperforms all baselines across Pass@{1,4,16,32,64,128}, with the advantage most pronounced under stricter limits. For example,

on AIME-24 with a 4000 cutoff, DLER-R1-7B achieves the highest Pass@1 accuracy and maintains its lead as K increases. Similar gains are observed on 1.5B models, where DLER-R1-1.5B surpasses both Laser-DE and LCR1. These results confirm that even under different test-time scaling budget, DLER preserves high accuracy, offering efficient and scalable reasoning across model sizes.

### 4.4 DLER ENABLES SUPERIOR TEST-TIME SCALING THROUGH PARALLEL THINKING

In the previous section, we showed that DLER models maintain superior accuracy across token budgets. Here, we extend the analysis to test-time scaling, benchmarking parallel thinking latency—the average elapsed time per question to generate multiple responses—using vLLM on a single NVIDIA H100 GPU with a 32k token cutoff. Evaluation is on AIME-24, reporting both Pass@K accuracy and per-question latency.

As shown in Fig. 1b, DLER-R1-7B achieves a single-response latency of 23.73s, down from 93.43s for DeepSeek-7B (4× faster). To reach 83.33% accuracy, it requires only 85.43s with 256 rollouts, compared to 221.22s with 16 rollouts on DeepSeek-7B (62% faster). Notably, generating 256 rollouts with DLER-7B remains slightly faster than producing a single response with DeepSeek-7B, while yielding 27% higher accuracy. Similar trends hold at the 1.5B scale as shown in Fig.8 and Table.4 in the appendix: DLER-R1-1.5B reduces single-response time from 58.99s (DeepSeek-1.5B) to 12.35s (4.8× faster), and reaches 80% accuracy in 52.09s with 128 rollouts, versus 229s for DeepSeek-1.5B with 64 rollouts (78% less time). Notably, generating 128 rollouts with DLER-1.5B remains faster than producing a single response with DeepSeek-1.5B, while yielding 50% higher accuracy. These results highlight a shift in perspective: **while prior work (Guo et al., 2025; Liu et al., 2025a) emphasized maximizing per-response accuracy with longer traces, our findings show that reasoning efficiency enables far greater test-time scaling.** In practice, allocating compute to efficient DLER models achieves higher accuracy within the same wall-clock time, making them far more practical for deployment.

### 4.5 DIFFERENT LENGTH PENALTIES NO LONGER PUSH THE ACCURACY–EFFICIENCY FRONTIER

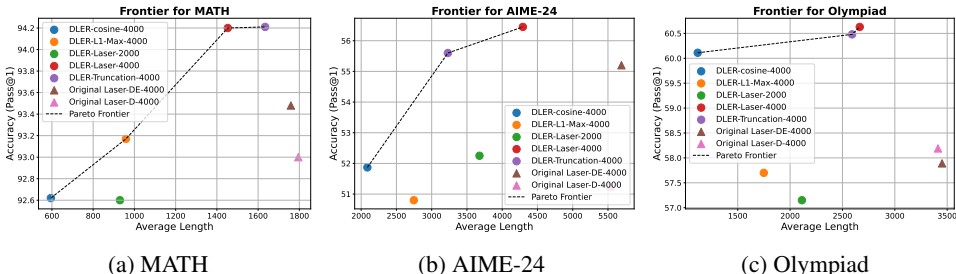

|     (a) MATH     |     (b) AIME-24     |     (c) Olympiad     |

Figure 6: Accuracy and average response length of DeepSeek-R1-7B trained using DLER with different length penalties on MATH and AIME-24. DLER establishes a new accuracy–length efficiency frontier, with varying length penalties moving performance along the frontier rather than beyond it.

With our optimization recipe (DLER), the effect of different length-penalty rewards changes fundamentally. Accuracy and length now form a trade-off rather than yielding shorter responses with higher accuracy. We test different length penalties with DLER: Truncation (zero reward beyond a cutoff), Cosine Liu et al. (2025a), L1-Max (Aggarwal & Welleck, 2025), and Laser (Liu et al., 2025b), denoted as "DLER-xxx-4000". Results for the original Laser-DE-4000 and Laser-D-4000 are based on the publicly released models from (Liu et al., 2025b).

As shown in Fig. 6, DLER consistently outperforms non-DLER counterparts. For example, DLER-Laser achieves higher accuracy than Original Laser-DE/D across tasks while maintaining shorter or similar lengths. In all cases, the accuracy–length Pareto frontier is defined entirely by DLER models, with different penalties shifting positions along the frontier rather than extending it. Importantly, the simplest length penalty—Truncation—remains highly competitive, often matching or surpassing more complex penalties like L1-Max and Laser. Truncation is also the most training-efficient since rollouts terminate at the cutoff, unlike L1-Max or Laser which require full sequences. This makes truncation not only a strong accuracy–efficiency option but also the most computationally practical.

### 4.6 OVERCOMING QUALITY LIMITATIONS OF PUBLICLY AVAILABLE TRAINING DATA

Table 2: Comparison of Llama-3.1-Nemotron-Nano-8B-v1 (Nemotron-8B), DLER-Llama-Nemotron-8B (DLER-Nemotron-8B) and the merged model **DLER-Nemotron-8B-Merge** (DLER-Nemotron-8B-Merge) in terms of Pass@1 accuracy and corresponding average output length (tokens) across benchmarks.

| Model | MATH ↑ | Length ↓ | AIME-24 ↑ | Length ↓ | AMC ↑ | Length ↓ | Minerva ↑ | Length ↓ | Olympiad ↑ | Length ↓ | Total Avg ↓ |
|---|---|---|---|---|---|---|---|---|---|---|---|
| Nemotron-8B | 95.40 | 3069 | 66.40 | 9899 | 88.25 | 6228 | 52.38 | 4031 | 64.33 | 6755 | 5996 |
| DLER-Nemotron-8B | 95.00 | **1843** | 63.54 | **3867** | 88.47 | **2850** | **54.27** | **2276** | **65.63** | 2843 | 2735 (-55%) |
| **DLER-Nemotron-8B-Merge** | **95.20** | 1995 | **66.66** | 5013 | **89.23** | 3358 | 53.19 | 2301 | 65.39 | 3520 | 3237 (-46%) |

Although reasoning models advance rapidly, their training datasets are rarely public, forcing practitioners to rely on easier public datasets that often under-challenge state-of-the-art models. To test this, we apply DLER—with a 6000-token truncation penalty and higher clipping threshold (0.36)—to Llama-3.1-Nemotron-Nano-8B-v1, a strong baseline surpassing DeepSeek-32B on MATH. The fine-tuned DLER-Nemotron-8B cuts average response length by 55% but suffers small accuracy drops on MATH (95.40 to 95) and AIME-24 (66.40 to 63.54), despite modest gains on AMC, Minerva, and Olympiad as shown in Table.2.

To recover accuracy without proprietary data, we explore weight merging. Since RL fine-tuning introduces sparse parameter updates (Mukherjee et al., 2025), merging the base and efficient models is promising. Naive approaches such as averaging or interpolation were unable to balance accuracy and length. Thus, inspired by (Yadav et al., 2023), we apply an update-selective method that retains the top 25% largest parameter deltas from the efficient model, rescales them by 0.7, and merges them into the original parameters. The merged model (DLER-Nemotron-8B-Merge) restores baseline accuracy on MATH (95.20) and AIME-24 (66.66), further boosts AMC, and maintains 46% shorter responses. Thus, update-selective merging provides a practical, training-free solution to recover accuracy while preserving efficiency when fine-tuning high-capacity models on public data.

## 5 RELATED WORK

Large reasoning models have inspired extensive research on improving efficiency. One line of work reduces reasoning through prompt design. For example, (Ma et al., 2025a) shows that explicit reasoning is often unnecessary in low-budget settings and proposes bypassing CoT with simple prompting. Another line focuses on supervised fine-tuning. (Lu et al., 2025) distill concise traces via an MCTS-inspired search, while (Chen et al., 2025) fine-tune with auxiliary verification to trigger self-reflection only when needed. Other approaches shorten reasoning by step-skipping (Liu et al., 2024), pruning unimportant tokens (Xia et al., 2025), or identifying parameter directions that directly control CoT length (Ma et al., 2025b). More recent work leverages reinforcement learning (RL) for principled accuracy–efficiency trade-offs. (Fang et al., 2025) use control tokens (<short>,<think>) with Decoupled GRPO to balance concise and detailed reasoning. (Liu et al., 2025b) present a unified reward-shaping framework, extending truncation with step-function penalties. (Luo et al., 2025a) encourage shorter reasoning by pre-estimating baseline performance before RL fine-tuning. (Hou et al., 2025) apply progressively tighter token limits across RL rounds, while (Aggarwal & Welleck, 2025) directly optimize both correctness and length constraints.

## 6 CONCLUSION

We revisited the problem of reasoning efficiency in large models by re-examining truncation, the simplest length penalty. Our analysis shows that prior accuracy loss stems not from penalty design but from unstable optimization. To address this, we propose *Doing Length pEnalty Right (DLER)*, which combines batch-wise reward normalization, higher clipping thresholds, dynamic sampling, and truncation. DLER achieves state-of-the-art accuracy–efficiency trade-offs, cutting response length by over 70% while fully recovering accuracy. We further introduce DA-DLER, which adapts truncation by question difficulty to push efficiency even further, and a weight merging strategy that restores accuracy when training data is limited, while halving output length. Overall, our results highlight that optimization, rather than complex penalty design, is the key to efficient and accurate reasoning models, opening new directions for scalable and accessible reasoning.

ETHICS STATEMENT

Our work builds on large language models (LLMs) and reinforcement learning for efficient reasoning. We do not collect or annotate any human subject data; all experiments use publicly available datasets under research licenses. We adhere to the terms of use specified by the original dataset creators and provide appropriate citations. Our approach does not introduce additional risks of data misuse or privacy leakage. Therefore, we do not foresee any obvious negative societal impacts. Nonetheless, we encourage responsible use and emphasize that our framework should be applied in alignment with safety and ethical guidelines.

REPRODUCIBILITY STATEMENT

We make every effort to ensure reproducibility of our results. Full implementation details and hyperparameter settings are provided in Section 4.1 and Appendix D. All datasets used in our experiments are publicly accessible and described in Sec. 4.1, along with the evaluation protocols and metrics. The codebase will be open-sourced upon acceptance.

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

## A  DLER ACHIEVES SOTA ACCURACY/LENGTH OF COT TRADE-OFF AND ENABLE BETTER TEST-TIME SCALING

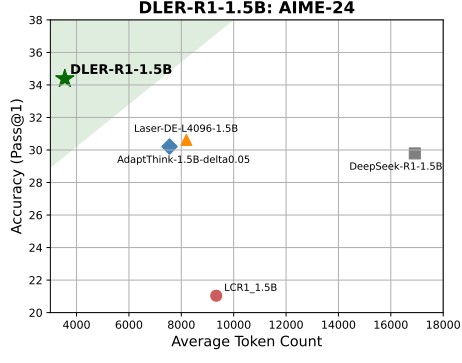

(a) DLER Training on DeepSeek-R1-1.5B

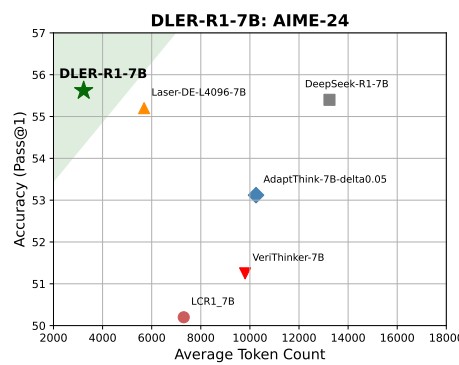

(b) DLER Training on DeepSeek-R1-7B

Figure 7: DLER achieves state-of-the-art Accuracy/Length of CoT trade-off. Compared to baseline models, DLER shortens the CoT by up to ∼70% while maintaining accuracy.

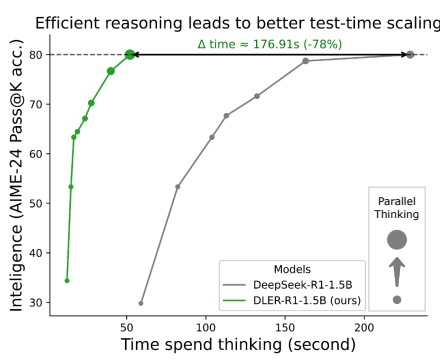

(a) DLER-R1-1.5B vs DeepSeek-R1-1.5B

(b) DLER-R1-7B vs DeepSeek-R1-7B

Figure 8: We test the test-time scaling on AIME-24 by varying the number of parallel rollouts (1-256) for DLER-R1 and DeepSeek-R1 models, using vLLM for benchmarking overall latency. DLER-R1 models demonstrate superior test-time scaling curves compared to DeepSeek-R1 models due to their improved concise reasoning ability.

# B LARGER REWARD VARIANCE RESULTS IN LARGER BIAS IN ADVANTAGE ESTIMATION.

## A.1 ASSUMPTIONS AND SETTINGS

We observe $N$ rewards $r_i$ for a prompt, and assume the true baseline is $\theta$, such that

$$r_i = \theta + \epsilon_i, \quad \epsilon_i \sim \mathcal{N}(0, \sigma^2), \quad i = 1, \dots, N,$$

with all advantage values $\epsilon_i$ independent. Define

$$\bar{\epsilon} = \frac{1}{N} \sum_{j=1}^{N} \epsilon_j, \quad D = \sqrt{\frac{1}{N} \sum_{j=1}^{N} (\epsilon_j - \bar{\epsilon})^2}, \quad A_i = \frac{\epsilon_i - \bar{\epsilon}}{D}.$$

We will first show that for any finite $N \geq 2$, the advantage estimator $A_i$ is biased:

$$\mathbb{E}[A_i \mid \epsilon_i] \neq \epsilon_i.$$

then we will show that given two different $\epsilon_i \sim \mathcal{N}(0, \sigma^2)$ and $\epsilon_i' \sim \mathcal{N}(0, \sigma'^2)$, if $\sigma' > \sigma$ then $\text{Bias}(\epsilon_i') > \text{Bias}(\epsilon_i)$ where $\text{Bias}(\epsilon_i) = \mathbb{E}[A_i \mid \epsilon_i]$.

Let us first derive why the advantage estimator $A_i$ is biased:

**Step 1: Bias in the Numerator.** The numerator can be expressed as

$$\epsilon_i - \bar{\epsilon} = \left(1 - \frac{1}{N}\right) \epsilon_i - \frac{1}{N} \sum_{j \neq i} \epsilon_j.$$

Since the $\epsilon_j$ with $j \neq i$ are zero-mean and independent of $\epsilon_i$, it follows that

$$\mathbb{E}[\epsilon_i - \bar{\epsilon} \mid \epsilon_i] = \left(1 - \frac{1}{N}\right) \epsilon_i.$$

**Step 2: Dependence of the Denominator on $\epsilon_i$.**

(a) Computing $\mathbb{E}[D^2 \mid \epsilon_i]$. By definition,

$$D^2 = \frac{1}{N} \sum_{j=1}^{N} (\epsilon_j - \bar{\epsilon})^2 = \frac{1}{N} \sum_{j=1}^{N} \epsilon_j^2 - \bar{\epsilon}^2.$$

Because

$$\bar{\epsilon} = \frac{1}{N} \left(\epsilon_i + \sum_{j \neq i} \epsilon_j\right),$$

and conditioning on $\epsilon_i$ leaves the $\epsilon_j$ (for $j \neq i$) as i.i.d. $\mathcal{N}(0, \sigma^2)$, we obtain

$$\mathbb{E}\left[\sum_{j=1}^{N} \epsilon_j^2 \mid \epsilon_i\right] = \epsilon_i^2 + (N-1)\sigma^2,$$

$$\mathbb{E}[\bar{\epsilon}^2 \mid \epsilon_i] = \frac{1}{N^2}\left(\epsilon_i^2 + (N-1)\sigma^2\right).$$

Thus,

$$\mathbb{E}[D^2 \mid \epsilon_i] = \frac{1}{N}\left(\epsilon_i^2 + (N-1)\sigma^2\right) - \frac{\epsilon_i^2 + (N-1)\sigma^2}{N^2} = \alpha + \beta\epsilon_i^2,$$

where $\alpha = \frac{(N-1)^2}{N^2}\sigma^2$ and $\beta = \frac{N-1}{N^2}$.

(b) Non-constancy of $g(\epsilon_i)$. Define

$$g(\epsilon_i) = \mathbb{E}\left[\frac{1}{D} \mid \epsilon_i\right], \qquad \mu(\epsilon_i) = \mathbb{E}[D^2 \mid \epsilon_i] = \alpha + \beta\epsilon_i^2.$$

Applying a Taylor expansion of $f(x) = x^{-1/2}$ around $x_0 = \mu(\epsilon_i)$ gives

$$f(x) \approx \frac{1}{\sqrt{x_0}} - \frac{1}{2}\frac{(x - x_0)}{x_0^{3/2}} + \frac{3}{8}\frac{(x - x_0)^2}{x_0^{5/2}} + O((x - x_0)^3).$$

Taking conditional expectation, we obtain

$$g(\epsilon_i) = \frac{1}{\sqrt{\mu(\epsilon_i)}} + \frac{3}{8}\frac{\mathrm{Var}(D^2 \mid \epsilon_i)}{\mu(\epsilon_i)^{5/2}}.$$

Since $\mu(\epsilon_i) = \alpha + \beta\epsilon_i^2$ with $\beta > 0$, $g(\epsilon_i)$ depends on $\epsilon_i^2$ and is therefore not constant.

**Step 3: Combining Results.** The estimator can be decomposed as

$$A_i = \frac{\epsilon_i - \bar{\epsilon}}{D} = \left(1 - \frac{1}{N}\right)\frac{\epsilon_i}{D} - \frac{1}{N}\left(\sum_{j \neq i}\epsilon_j\right)\frac{1}{D}.$$

For fixed $\epsilon_i$, the distribution of $\sum_{j \neq i}\epsilon_j$ is symmetric about zero, while $1/D$ is always positive. Hence

$$\mathbb{E}\left[-\frac{1}{N}\sum_{j \neq i}\epsilon_j \cdot \frac{1}{D} \,\Big|\, \epsilon_i\right] = 0.$$

It follows that

$$\mathbb{E}[A_i \mid \epsilon_i] = \left(1 - \tfrac{1}{N}\right)\epsilon_i \cdot g(\epsilon_i).$$

**Step 4: Concluding the Bias.** If $A_i$ were unbiased, we would require

$$\left(1 - \tfrac{1}{N}\right)g(\epsilon_i) \equiv 1 \quad \Rightarrow \quad g(\epsilon_i) \equiv \frac{N}{N - 1}.$$

This contradicts Step 2, which showed $g(\epsilon_i)$ depends on $\epsilon_i^2$. Therefore, for any finite $N \geq 2$,

$$\mathbb{E}[A_i \mid \epsilon_i] \neq \epsilon_i.$$

Hence, $A_i$ is a biased estimator.

Next, we will show that given two different $\epsilon_i \sim \mathcal{N}(0, \sigma^2)$ and $\epsilon_i' \sim \mathcal{N}(0, \sigma'^2)$, if $\sigma' > \sigma$ then $\mathrm{Bias}(\epsilon_i') > \mathrm{Bias}(\epsilon_i)$ where $\mathrm{Bias}(\epsilon_i) = \mathbb{E}[A_i \mid \epsilon_i]$.

**Conditional bias (exact formula)**

From Step 3, we know that the conditional expectation is

$$\mathbb{E}[A_i \mid \epsilon_i] = \left(1 - \frac{1}{N}\right)\epsilon_i\, g(\epsilon_i), \qquad g(\epsilon_i) = \frac{1}{\sqrt{\mu(\epsilon_i)}} + \frac{3}{8}\frac{\mathrm{Var}(D^2 \mid \epsilon_i)}{\mu(\epsilon_i)^{5/2}}$$

Hence the **bias** is

$$\mathrm{Bias}(\epsilon_i) = \mathbb{E}[A_i \mid \epsilon_i] - \epsilon_i = \epsilon_i\left[\left(1 - \frac{1}{N}\right)g(\epsilon_i) - 1\right].$$

Since $\sigma \propto g(\epsilon_i)$ and $g(\epsilon_i) \propto \mathrm{Bias}(\epsilon_i)$, we deduce that if $\sigma' > \sigma$, then $\mathrm{Bias}(\epsilon_i') > \mathrm{Bias}(\epsilon_i)$.

## A.2 GROUP-WISE REWARD NORMALIZATION (GRPO) AND BATCH-WISE NORMALIZATION

Figure 9: Accuracy and average response length of DeepSeek-R1-7B on the AIME-24 test set, evaluated every 10 training steps across two RL training runs: group-wise reward normalization (GRPO) and batch-wise normalization. GRPO shows declining accuracy while batch-wise reward normalization remains stable despite reduced token counts.

# C   COMBINING ALL INGREDIENTS: DO LENGTH PENALTY RIGHT

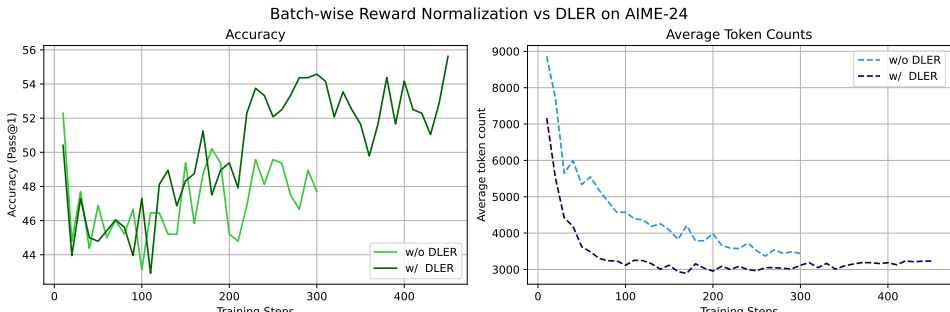

Figure 10: Accuracy and average response length of DeepSeek-R1-7B on the AIME-24 test set, evaluated every 10 training steps across two RL training runs: one with batch-wise reward normalization and the other with our DLER recipe. DLER enhances the accuracy–token efficiency by reducing token usage while fully recovering the accuracy loss of applying plain batch-wise reward normalization.

## D  HYPERPARAMETERS SETTING

Table 3: DLER veRL training configuration

| Parameter | Value |
|---|---|
| data.train_batch_size | 512 |
| actor_rollout_ref.actor.ppo_mini_batch_size | 64 |
| actor_rollout_ref.actor.ppo_epochs | 1 |
| data.max_prompt_length | 1024 |
| actor_rollout_ref.actor.optim.lr | 1.00E-06 |
| actor_rollout_ref.rollout.temperature | 1 |
| actor_rollout_ref.rollout.n | 16 |
| actor_rollout_ref.actor.clip_ratio_low | 0.2 |
| actor_rollout_ref.actor.clip_ratio_high | 0.28 |
| algorithm.filter_groups.enable | TRUE |
| algorithm.filter_groups.metric | seq_reward |
| actor_rollout_ref.actor.kl_loss_coef | 0.0005 |
| actor_rollout_ref.actor.kl_loss_type | mse |

972
973
974
975
976
977
978
979
980
981
982
983
984
985
986
987
988
989
990
991
992
993
994
995
996
997
998
999
1000
1001
1002
1003
1004
1005

# E  PERFORMANCE UNDER DIFFERENT TEST-TIME SCALING SETTINGS

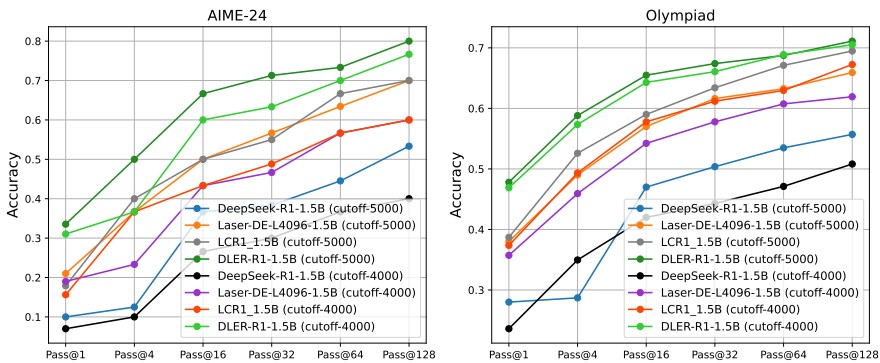

(a) DLER-R1-1.5B vs Baselines

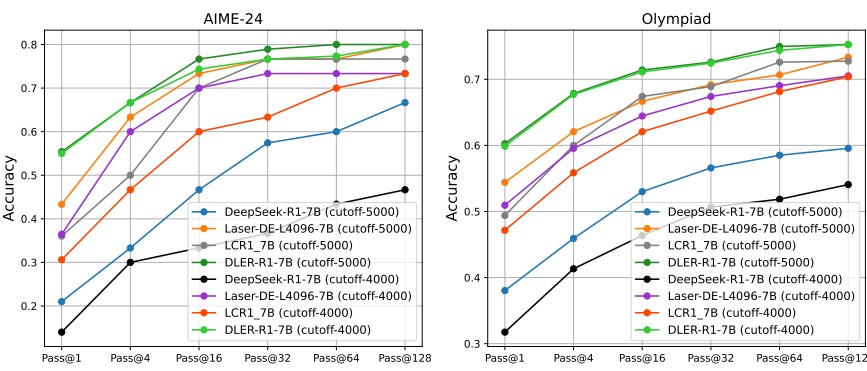

(b) DLER-R1-7B vs Baselines

Figure 11: Pass@K accuracy of DLER-1.5B/7B versus baselines on AIME-24 and Olympiad under different length cutoffs. DLER consistently outperforms all baselines across different test-time budget settings.

# F    PARALLEL THINKING LATENCY

Table 4: Average Parallel Inference Latency per Request: DeepSeek-R1-7B vs DLER-R1-7B

|  | #Parallel Thinking | Accuracy | Avg Request (Sec) |
|---|---|---|---|
| DeepSeek-R1-1.5B | 1 | 29.79 | 58.99 |
|  | 4 | 53.33 | 82.26 |
|  | 8 | 63.33 | 103.86 |
|  | 12 | 67.68 | 112.96 |
|  | 16 | 71.62 | 132.30 |
|  | 32 | 78.72 | 163.00 |
|  | 64 | 80.00 | 229.00 |
| DLER-R1-1.5B | 1 | 34.37 | 12.35 |
|  | 4 | 53.33 | 14.83 |
|  | 8 | 63.33 | 16.64 |
|  | 12 | 64.44 | 18.91 |
|  | 16 | 67.10 | 23.70 |
|  | 32 | 70.24 | 27.64 |
|  | 64 | 76.67 | 39.99 |
|  | 128 | 80.00 | **52.09** |
| DeepSeek-R1-7B | 1 | 55.40 | 93.43 |
|  | 4 | 77.75 | 144.44 |
|  | 8 | 81.28 | 174.17 |
|  | 12 | 82.47 | 200.51 |
|  | 16 | 83.33 | 221.22 |
| DLER-R1-7B | 1 | 55.60 | 23.73 |
|  | 4 | 66.67 | 26.89 |
|  | 8 | 73.20 | 29.17 |
|  | 12 | 73.88 | 32.45 |
|  | 16 | 75.17 | 34.44 |
|  | 32 | 77.69 | 43.83 |
|  | 64 | 79.56 | 55.43 |
|  | 128 | 81.67 | 57.02 |
|  | 256 | 83.33 | **85.43** |

# G ANALYSIS

## G.1 ENTROPY DISTRIBUTION

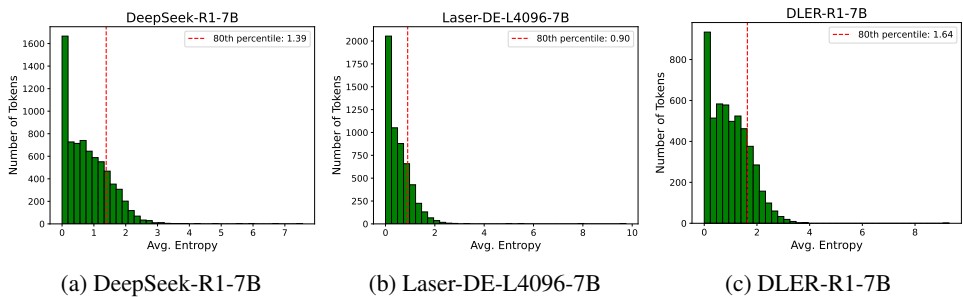

(a) DeepSeek-R1-7B      (b) Laser-DE-L4096-7B      (c) DLER-R1-7B

Figure 12: Token entropy distribution of DeepSeek-R1-7B, Laser-DE-L4096-7B, and DLER-R1-7B on AIME-24. Laser-DE-L4096-7B shows a markedly contracted distribution relative to DeepSeek-R1-7B, indicating fewer high-entropy tokens and reduced reasoning exploration capability, while DLER-R1-7B exhibits a slight increase in such tokens.

In this section, we follow the setup of Wang et al. (2025) to examine the distribution of generation entropy in chain-of-thought reasoning at the token level to evaluate how reasoning compression affects model diversity and exploration ability. We evaluate three models: the original DeepSeek-R1-7B, the released Laser-DE-L4096-7B from Liu et al. (2025b), and our DLER-R1-7B, generating responses for AIME-24 questions with 16 rollouts per question. Token-level entropy is computed following the procedure in Wang et al. (2025). Across all three models, regardless of whether they have undergone efficient reasoning RL training, we observe a consistent pattern: only a small fraction of tokens exhibit relatively high entropy, while the majority have low entropy. This results in a right-skewed token entropy distribution for all models, consistent with the findings in Wang et al. (2025). However, we also find notable differences post the efficient RL training. The entropy distribution of Laser-DE-L4096-7B contracts significantly, indicating a reduction in the number of high-entropy tokens, whereas DLER-R1-7B shows a slight increase in such tokens. As noted in Wang et al. (2025) and our analysis in Sec. 3.2, high-entropy tokens are typically those that initiate exploration and reasoning steps. Therefore, the observed decrease in high-entropy tokens for Laser-DE-L4096-7B suggests diminished exploration capacity, while the modest increase in DLER-R1-7B indicates that our training recipe better preserves this ability. This preservation of exploration contributes to DLER-R1-7B's superior accuracy and shorter response lengths after RL training.

## G.2 REASONING TRACE ANALYSIS

Table 5: Average tokens per step, total steps, and count of transition keywords per response for DeepSeek-R1-7B, Laser-DE-L4096-7B, and DLER-R1-7B on the AIME-24.

| Model | #Tokens per step | | | #Steps | | | #Keywords | | |
|---|---|---|---|---|---|---|---|---|---|
| | Overall | Correct | Incorrect | Overall | Correct | Incorrect | Overall | Correct | Incorrect |
| DeepSeek-R1-7B | 34 | 34 | 34 | 461 | 245 | 736 | 207 | 85 | 361 |
| Laser-DE-L4096-7B | 37 | 35 | 38 | 175 | 122 | 240 | 82 | 35 | 140 |
| **DLER-R1-7B** | 29 | 29 | 30 | 118 | 108 | **131 (-83%)** | 51 | 28 | **81 (-78%)** |

In this section, we analyze the reasoning trajectory to assess the impact of our proposed training recipe on mitigating the overthinking problem in reasoning models. Using the same generation setup on AIME-24 as in the previous entropy distribution analysis, reasoning steps are segmented by double newline (\n \n) delimiters in the model's output. Following the definition of reasoning keywords in Lu et al. (2025), we designate the following terms as keywords: {'But', 'Wait', 'Alternatively', 'However', 'Hmm', 'Hmmm', 'Not sure', 'Going back', 'Backtrack', 'Trace back', 'Another'}.

Table 5 reports reasoning trace statistics on AIME-24 for DeepSeek-R1-7B, Laser-DE-L4096-7B, and DLER-R1-7B. DLER-R1-7B shows a marked reduction in average reasoning steps compared to both baselines, with the most notable improvement in incorrect responses—achieving the usage of only 131 reasoning steps, a 45% decrease relative to Laser-DE and an 83% decrease relative to the original model. A similar trend is observed for reasoning keywords, where DLER-R1-7B yields the fewest overall and the lowest number in incorrect cases. As prior works Chen et al. (2025); Lu et al. (2025); Liu et al. (2025b) have discovered, overthinking is a prevalent phenomenon, particularly when facing challenging questions or uncertainty, often leading to unnecessarily prolonged reasoning or or even near-infinite loops when no final answer is produced. The observed reductions in reasoning steps and keywords demonstrate that our method effectively curtails overthinking, resulting in more concise reasoning trajectories and thereby improving both efficiency and accuracy.

## H    USE OF LLMS

In this work, LLMs were used solely for grammatical editing. The research ideas, methodology, experimental design, and analyses were were entirely conducted by the authors.

