# OpenReview forum: "DLER: Doing Length pEnalty Right — Incentivizing More Intelligence per Token via Reinforcement Learning"
_ICLR.cc/2026/Conference — Submitted to ICLR 2026_

### Official Review · Reviewer_wEUt · 2025-10-16

**Soundness:** 2
**Presentation:** 2
**Contribution:** 1
**Rating:** 4
**Confidence:** 4

**Summary:**

This manuscript focuses on improving the reasoning efficiency of large reasoning models. The authors identify 3 major challenges including biased advantage estimation, entropy collapse and sparse rewards. To this end, the authors propose DLER, a RL training recipe for efficient reasoning capability. DLER consists a suite of tricks including reward shaping for length penalty, dynamic sampling, clip high and batch normalization for advantage estimation. Experiments on DeepSeek-R1-1.5B/7B and mathmetical benchmarks are carried out to evaluate the proposed training recipe.

**Strengths:**

- Reasoning efficiency is a widely acknowledged common challenge for large reasoning models. Thus the manuscript is well motivated and highly in-time.
- Most parts of this manuscripts are easy to follow up with. The organization and writting is clear.

**Weaknesses:**

- Novelty and contribution. From my reading of this manuscript, the contribution and novelty of DLER is hard for me to recognize. For instance, the majority of DLER, i.e. dynamic sampling, batch normalization of rewards, and clip high are proposed by DAPO. To my best knowledge, DAPO has been acccepted by RL community as the most commonly used baseline. Thus the novelty and contribution of DLER is very limited from my view. The manuscript seems to be a reproduction report of DAPO rather than a technical paper for possible publication on ICLR.

- Continue with my last point, the only newly proposed trick of DLER seems to be the truncation length penalty. However, a soft length penalty is also adopted in DAPO. There is no direct comparison between DLER and DAPO in the experiment section. I doubt if DLER can beat DAPO.
- Many other ``findings'' in this manucript are also not new for the reviewer. For example, there has been many previous works try to improve reasoning efficiency by model merging.
- The experiments are only conducted on DeepSeek-R1-1.5B/7B, which is relatively limited. The reviewer suggest more comparison on other model family such as Llama, OctoThinker or Mistral.
- The resulted DLER-R1-1.5B is trained on the same dataset with DeepScaleR-1.5B-Preview. However, the AIME24 accuracy is noticeably lower than DeepScaleR-1.5B-Preview (of which the pass@1 accuracy is 43.1), which raising further concerns about the effectiveness of DLER as well as limited baselines.

**Questions:**

- Could the authors further explain how the proposed DLER differ from DAPO in performance or method level?
- Could the authors compare DLER with DeepScaleR-1.5B-preview as these two models are trained on the same dataset.

---

> ### Author Response · Authors · 2025-11-21
>
> Thank you for reviewing our work and for your thoughtful questions. Our detailed responses are provided below.
>
> **[Q1: Could the authors further explain how the proposed DLER differs from DAPO in performance or method level?]**
>
> Conceptually, DLER differs from DAPO in terms of 1) how to perform normalization (Batchwise vs Groupwise) and 2) which length penalty is used (truncation vs soft overlength penalty) and to provide a direct performance comparison, we trained DeepSeek-R1-7B using the DAPO algorithm.  We refer to this model as DAPO-7B. DAPO-7B uses the same hyperparameters, same target length (4000), and the same number of training steps as DLER. Below we compare DAPO-7B with the original DeepSeek-R1-7B and our DLER-R1-7B:
>
> | Model          | Math 500 | Length | AIME  | Length | AMC   | Length | Minerva | Length | Olympiad | Length | Avg Acc. | Avg Length |
> |----------------|----------|--------|-------|--------|-------|--------|---------|--------|----------|--------|----------|------------|
> | DeepSeek-R1-7B | 93.60    | 3999   | 55.40 | 13241  | 82.90 | 7461   | 49.79   | 5199   | 58.21    | 8837   | 67.98    | 7747       |
> | DAPO-7B     | 93.89    | 1743   | 54.12 | 3764   | 83.74 | 2417   | 50.63   | 2245   | 58.40    | 3109   | 68.16    | 2656       |
> |  **DLER-R1-7B**     | **94.21**   | **1634**   | **55.62** |  **3230**  | **84.41** | **2512**   | **53.88**   | **2058**   | **60.48**    | **2592**   | **69.72**    | **2405**      |
>
> From the results, we can see that across all benchmarks, DLER-R1-7B consistently outperforms DAPO-7B—achieving both higher average accuracy and shorter average response length. This highlights the effectiveness of our training recipe and indicates that each component of DLER plays an important role in resolving the length penalty related issues identified in our analysis.
>
> **[Q2: What is the novelty of the paper]**
>
> Regarding the concern about novelty, while we agree that the individual components of our method draw on prior work, how to deploy these techniques together to achieve effective CoT length reduction without performance degradation is unclear and turns out to be non-trivial in our exploration. Thus, our main contributions lie in (1) providing the first systematic analysis of the optimization challenges that arise when applying length penalties to reduce COT length, (2) showing how to resolve these issues by carefully combining existing techniques into a principled and effective training recipe, and (3) demonstrating that the choice of optimization techniques is far more important than designing increasingly complex length rewards, (4) open the black box of why and how existing approaches improve training dynamics from the root.
>
> Specifically, prior studies on efficient reasoning focus on designing increasingly complex reward functions which they overlook the underlying optimization instability that arises when applying those length rewards. Our paper identifies this optimization bottleneck clearly and shows why naïvely applying length penalties with default optimization setups leads to degradation in performance.
>
> Building on this analysis, we curate and adapt techniques from DAPO and Hu et al. (2025), integrating them into DLER, a training recipe that achieves state of the art accuracy/length tradeoffs using the simplest possible length reward. Our work demonstrates that more complex designs of length penalties are not necessary: what matters is selecting the right optimization strategies and combining them coherently.

---

> ### Author Response · Authors · 2025-11-21
>
> **[Q3: More experiments on other models than DeepSeek-R1-1.5B/7B]**
>
> We have already included additional experiments on NVIDIA’s nvidia/Llama-3.1-Nemotron-Nano-8B-v1 model in Section 4.6. These results demonstrate that DLER reduces average response length by 55% while improving accuracy on most tasks, with only a small drop on AIME. For practitioners seeking strictly lossless accuracy across all benchmarks, we also provide a model merging approach that trades a small portion of the length reduction for full accuracy retention, achieving a 46% decrease instead.
>
> | Model                     | MATH  | Length  | AIME-24  | Length | AMC  | Length  | Minerva  | Length  | Olympiad | Length | Avg Length |
> |---------------------------|--------|-----------|------------|-----------|--------|-----------|------------|------------|-------------|------------|--------------|
> | Nemotron-8B               | 95.40  | 3069      | 66.40     | 9899      | 88.25 | 6228      | 52.38     | 4031       | 64.33       | 6755       | 5996         |
> | DLER-Nemotron-8B          | 95.00  | **1843**  | 63.54     | **3867**  | 88.47 | **2850**  | **54.27** | **2276**   | **65.63**   | **2843**   | **2735 (-55%)** |
> | **DLER-Nemotron-8B-Merge** | **95.20** | 1995 | **66.66** | 5013 | **89.23** | 3358 | 53.19 | 2301 | 65.39 | 3520 | **3237 (-46%)** |
>
> **[Q4: Could the authors compare DLER with DeepScaleR-1.5B-preview as these two models are trained on the same dataset?]**
>
> To address this question, we trained a DLER variant of DeepScaleR-1.5B using the truncation-based penalty (target length 4000 tokens) for 400 steps, with the same hyperparameters and dataset used for training DeepSeek-R1-7B under DLER. This allows us to directly assess whether DLER can still improve reasoning efficiency on a model that has already undergone extensive RL post training.
>
> | Model                 | Math 500 | Length | AIME  | Length | AMC   | Length | Minerva | Length | Olympiad | Length | Avg Acc. | Avg Length |
> |-----------------------|----------|------------|-------|----------|-------|----------|---------|----------|----------|----------|----------|---------------|
> | Deepscaler-1.5B       | 89.22    | 3151       | 38.12 | 9671     | 73.04 | 5676     | 42.00   | 4954     | 50.80    | 5912     | 58.60    | 5873          |
> |  **Deepscaler-DLER-1.5B**  | 87.95    | **1631**       | 37.78 | **2959**     | **74.24** | **2282**     | **44.48**   | **2025**     | 49.75    | **2270**     | **58.80**    | **2233**   |
>
> From these results, we see that DLER reduces the average response length by 62% while maintaining comparable accuracy which demonstrates that DLER remains effective even for models that have already been heavily trained with RL. Finally, we notice that when evaluating the base DeepScaleR-1.5B model with our evaluation codebase, we were unable to reproduce the reported AIME pass@1 accuracy of 43.1. To ensure consistent performance comparison, we therefore report results obtained using our evaluation pipeline, which computes pass@1 by averaging across 16 generated samples.

---

### Official Review · Reviewer_nZtF · 2025-10-29

**Soundness:** 2
**Presentation:** 3
**Contribution:** 1
**Rating:** 4
**Confidence:** 3

**Summary:**

This paper presents an integrated method to enhance reasoning efficiency in LLMs by focusing on the length of reasoning outputs. Existing long-to-short methods through RL often degrade accuracy due to suboptimal optimization strategies. The authors propose DLER, which optimizes RL with simple truncation penalties and batch-wise reward normalization, mitigating issues like bias in advantage estimation. Other widely adopted techniques such as clipping higher and dynamic sampling are also integrated. They demonstrate that DLER cuts output length by over 70% and achieves highest accuracy over other length penalty baselines.

**Strengths:**

1. **Well-considered integration:** The framework integrates length penalty, clipping regions, and dynamic sampling techniques, solving existing issues in RL.
2. **Empirical significance and extensive experimentation:** Significant reductions in reasoning length (up to 70%) while maintaining or improving accuracy.
3. **Analysis on different length penalties:** Ablation studies include replacing different length penalties are clear and thorough.

**Weaknesses:**

1. **Lack of focus:** The method integrates a bundle of existing techniques in long-to-short RL without explicit focus. The analysis of different length penalties is most contributory to me, but is not the focus of the paper.
2. **Experimental result interpretation:** Some claims are at least exaggerated from the emprical results. For instance, the experiments in Sec.4.5 only show the different trade-offs and  some of the length penalty do have better "per token performance" than truncation; more rigorous experiments including adjusting differnt truncation lengths and penalty strengths are required for the current claim. Also Sec.4.6 is confusing since DLER does not lose much performance compared to the base model.
3. **Limited scope:** The paper only considers math problems and does not discuss further generalization.

**Questions:**

1. Why is Fig.4(c) has similar avg. response lengths at step 0 and step 400? And how do the savings of reasoning tokens establish under this circumstance?
2. How do you envision the objective of this area? Should the per-token performance or the best accuracy under whatever cost reduction be the focus?

---

> ### Author Response · Authors · 2025-11-21
>
> Thank you for reviewing our work and for your thoughtful questions. Our detailed responses are provided below.
>
> **[Q1: Lack of focus]**
>
> The focus of this work is to improve RL training when length penalties are applied. We begin by explaining why this focus is important yet challenging by showing why prior research has largely concentrated on designing increasingly complex reward functions due to the belief that simple truncation length penalty is ineffective. Our analysis challenges this assumption. We show that the problem does not stem from the simplicity of truncation, but from the fact that it has been used without the appropriate optimization techniques. When paired with the correct training recipe, even the simplest truncation penalty can achieve SOTA results.
>
> We then demonstrate how to derive a right training algorithm. We start by systematically analyzing the underlying issues that prevent length penalties from working well under standard RL setups and identify how to address them. Based on these insights, we propose a training recipe that assembles the right optimization techniques, achieving SOTA accuracy/efficiency tradeoffs. Moreover, with DLER, the specific choice of length penalty becomes far less pivotal and different length penalties no longer shift the accuracy/efficiency frontier individually, but together define a new Pareto frontier.
>
> **[Q2: Confusion regarding Sec.4.5, Sec.4.6, and Fig.4(c)]**
>
> For Sec.4.5, we do not claim that truncation-based length penalties are universally optimal. Rather, our results show that, when combined with the proper training recipe, different length penalties fall on the new Pareto frontier that surpasses prior methods which rely on the default GRPO recipe. We further demonstrate that DLER-truncation can match or even outperform more complex penalties like L1-Max and Laser under a fixed cost budget. This makes truncation not only a strong choice in terms of the accuracy/efficiency tradeoff, but also the cheapest option to train, since rollouts can be stopped exactly at the target length which something other length penalty functions do not allow.
>
> Regarding confusion about Section 4.6, we agree that DLER already preserves most of the performance of the base model. The merging method is introduced as a practical tool that allows practitioners to quickly adjust the accuracy/length tradeoffs without retraining, if they can’t take any accuracy loss.
>
> The confusion about Fig. 4(c), which shows similar average response lengths at step 0 and step 4000, is due to the use of a truncation length penalty. Applying truncation length penalty only allows the model’s maximum response length to be 4000 tokens throughout training, regardless of the actual COT length. As a result, even though the true average response length is about 10000 tokens at step 0, the average response length after applying truncation can only be under the 4000 token cap. We will add this explanation to the manuscript to avoid confusion. Thanks for bringing this up.
>
> [**Q3: DLER is only conducted on math reasoning tasks, can it be generalized to other tasks?]**
>
> To evaluate the generalizability of DLER beyond math reasoning, we conducted additional experiments on coding reasoning tasks. Specifically, we trained DeepSeek-R1-7B using the PRIME-RL/Eurus-2-RL dataset, a high-quality RL dataset consisting of programming problems. We applied the same DLER recipe used for math reasoning, including identical hyperparameters, and set the target length for truncation to 5000 tokens. Training was run for 400 steps.
>
> We then evaluated the resulting model under the same decoding settings described in the paper (temperature, top-p, max length) across four coding benchmarks: APPS [1], CodeContests [2], Codeforces [3], and TACO [4]. We report pass@1 accuracy and average response length.
>
> | Model                          | Apps  | Length | Codecontests | Length | Codeforces | Length | Taco  | Length | Avg Acc | Avg Length |
> |--------------------------------|-------|--------|--------------|--------|------------|--------|-------|--------|---------|------------|
> | DeepSeek-R1-7B    | 28.07 | 9474   | 47.25        | 11326  | 46.49      | 11136  | 28.07 | 10316  | 37.47   | 10563      |
> |  **DLER-7B-Code**                          | **62.39** | **2770**   | **58.85**        | **3128**   | **62.12**      | **3074**   | **37.77** | **2916**   | **55.28 (+17.8%)**     |  **2972 (-72%)**   |
>
> The results show that DLER remains highly effective in coding reasoning tasks where it reduces average response length of DeepSeek-R1-7B by 71% while improving accuracy by 17.8% across the four datasets. This demonstrates that DLER is not limited to math reasoning and can generalize to other forms of complex reasoning, including code generation tasks.

---

> ### Author Response · Authors · 2025-11-21
>
> **[Q4:  How do you envision the objective of this area? Should the per-token performance or the best accuracy under whatever cost reduction be the focus?]**
>
> We believe there is no single answer to your question, as the appropriate objective depends on the different use cases. If a user wants to enforce a fixed length budget per response, then evaluating methods based on the best accuracy achievable within that target length is most meaningful. Conversely, if a user does not face strict length limits and primarily aims to improve overall reasoning efficiency, then prioritizing per-token accuracy becomes more appropriate.
>
>
> [1] Measuring coding challenge competence with apps, 2021
>
> [2]  Competition-level code generation with alphacode, 2022
>
> [3] https://huggingface.co/datasets/MatrixStudio/Codeforces-Python-Submissions
>
> [4] Taco: Topics in algorithmic code generation dataset. 2023

---

### Official Review · Reviewer_YCru · 2025-10-31

**Soundness:** 3
**Presentation:** 3
**Contribution:** 3
**Rating:** 8
**Confidence:** 4

**Summary:**

The paper revisits reinforcement learning for length-efficient reasoning in LLMs. It argues that performance drops under length penalties come from suboptimal optimization rather than the penalty itself. Using simple truncation, the authors identify three causes—biased advantage estimation, entropy collapse, and sparse reward signals—and propose DLER, which combines batch-wise normalization, higher clipping, dynamic sampling, and truncation. DLER shortens responses by about 70% while maintaining or improving accuracy across math reasoning benchmarks. A difficulty-aware variant further reduces length, and a weight-merging method restores accuracy when training data are limited.

**Strengths:**

- Clear diagnosis of why RL with truncation fails and how each fix addresses it.
- Consistent gains across multiple datasets and model sizes.
- Useful weight-merging strategy for limited-data scenarios.

**Weaknesses:**

- Evaluations limited to math reasoning; unclear generalization to other non-math reasoning tasks.
- Difficulty-aware truncation may bias the training distribution toward easier samples, reducing exposure to genuinely hard problems that need longer reasoning.

**Questions:**

- What is the results of DLER on non-math or multimodal reasoning tasks?
- Does the normalization or dynamic sampling amplify certain reward trends?
- Could the difficulty-aware mechanism introduce potential biases or unintended drawbacks?

---

> ### Author Response · Authors · 2025-11-21
>
> Thank you for reviewing our work and for your thoughtful questions. Our detailed responses are provided below.
>
> **[Q1: What are the results of DLER on non-math or multimodal reasoning tasks?]**
>
> To evaluate the generalizability of DLER beyond math reasoning, we conducted additional experiments on coding reasoning tasks. Specifically, we trained DeepSeek-R1-7B using the PRIME-RL/Eurus-2-RL dataset, a high-quality RL dataset consisting of programming problems. We applied the same DLER recipe used for math reasoning, including identical hyperparameters, and set the target length for truncation to 5000 tokens. Training was run for 400 steps.
>
> We then evaluated the resulting model under the same decoding settings described in the paper (temperature, top-p, max length) across four coding benchmarks: APPS [1], CodeContests [2], Codeforces [3], and TACO [4]. We report pass@1 accuracy and average response length.
>
> | Model                          | Apps  | Length | Codecontests | Length | Codeforces | Length | Taco  | Length | Avg Acc | Avg Length |
> |--------------------------------|-------|--------|--------------|--------|------------|--------|-------|--------|---------|------------|
> | DeepSeek-R1-7B    | 28.07 | 9474   | 47.25        | 11326  | 46.49      | 11136  | 28.07 | 10316  | 37.47   | 10563      |
> |  **DLER-7B-Code**                          | **62.39** | **2770**   | **58.85**        | **3128**   | **62.12**      | **3074**   | **37.77** | **2916**   | **55.28 (+17.8%)**     |  **2972 (-72%)**   |
>
> The results show that DLER remains highly effective in coding reasoning tasks where it reduces average response length of DeepSeek-R1-7B by 71% while improving accuracy by 17.8% across the four datasets. This demonstrates that DLER is not limited to math reasoning and can generalize to other forms of complex reasoning, including code generation tasks.
>
> **[Q2: Does normalization or dynamic sampling amplify certain reward trends?]**
>
> We interpret this question as referring to how the two techniques influence reward and response length trends during training. However, if you have a different question in mind, please let us know so we can address it further.
>
> In our experiments, replacing groupwise normalization with batchwise normalization resulted in a noticeably more stable correctness reward curve. Under batchwise normalization, the reward signal increases steadily throughout training and reaches a higher plateau (0.73 vs. 0.68). In contrast, groupwise normalization produces a more volatile reward trend that saturates earlier at a lower score. This aligns with our analysis that batchwise normalization reduces reward variance and thus provides a more stable optimization signal.
>
> Regarding dynamic sampling, we observed that applying it leads to consistently longer average response lengths during training. Specifically, with dynamic sampling, the average batch response length plateaus around 2400 tokens, whereas without dynamic sampling it plateaus around 2000 tokens, even with a 4,000 token target length. This indicates that dynamic sampling helps prevent overfitting to easy prompts, which would otherwise cause the model to collapse to shorter reasoning traces prematurely.
>
> We will add the visualization of those reward trends to the manuscript.
>
> **[Q3: Could the difficulty-aware mechanism introduce potential biases or unintended drawbacks?]**
>
> Conceptually, the difficulty-aware truncation method shouldn't introduce potential bias because each sample’s target length is continuously adjusted according to how well the model can answer that question which is measured by its average pass rate across rollouts. As the model becomes better toward the latter stage of training, the target length tightens, preventing it from overfitting to problems it has already learned to solve under more lenient length limits.
> Empirically, we did not observe noticeable decrease of accuracy after applying the difficulty-aware mechanism, for instance, DA-DLER-R1-1.5B maintain the same AIME pass@1 accuracy of 34.3 as DLER-R1-1.5B and even outperform DLER-R1-1.5B on AMD/Minerva/Olympiad by 1.88/1.3/0.39 respectively while producing shorter responses on average.
>
> [1] Measuring coding challenge competence with apps, 2021
>
> [2]  Competition-level code generation with alphacode, 2022
>
> [3] https://huggingface.co/datasets/MatrixStudio/Codeforces-Python-Submissions
>
> [4] Taco: Topics in algorithmic code generation dataset. 2023

---

### Official Review · Reviewer_B7sj · 2025-11-01

**Soundness:** 3
**Presentation:** 3
**Contribution:** 2
**Rating:** 4
**Confidence:** 4

**Summary:**

The paper proposes DLER, a training recipe for RL-based reasoning models aimed at shortening solution traces while maintaining accuracy. The recipe pairs a truncation-style length penalty with three stabilizers: batch-wise advantage normalization, a higher clipping threshold intended to preserve gradients for exploratory tokens, and dynamic sampling to avoid degenerate reward batches. Experiments on math-reasoning benchmarks report competitive accuracy with noticeably more concise reasoning trace and improved test-time efficiency.

**Strengths:**

- The empirical setup is described in a way that allows a reader to trace hyperparameters, datasets, and evaluation choices end to end. Ablations and reporting are organized so the effect of each component is inspectable. This clarity materially improves reproducibility and makes it easier to audit design choices and replicate results.
- The produced models achieve solid reasoning performance while reducing unnecessary verbosity in the generated traces. The improvements are presented consistently across tasks, indicating that the gains are not limited to a single benchmark. In short, the evidence suggests the method yields better performance per token without obvious accuracy collapse.

**Weaknesses:**

I appreciate the authors revisit a few techniques utilized in prior papers and package them into an effective recipe for training a reasoning model. However, the paper’s distinct technical contribution is hard to identify: the clip-high and dynamic sampling components follow DAPO, and batch-wise advantage normalization appears in Hu et al. (2025). In this work these elements largely behave as expected and are not clearly repurposed or extended for a new objective. Overall, this reads as a strong engineering consolidation rather than a substantive research advance.

**Questions:**

see weaknesses

---

> ### Author Response · Authors · 2025-11-21
>
> Thank you for reviewing our work and for your thoughtful questions. Our detailed responses are provided below.
>
> Regarding the concern about novelty, while we agree that the individual components of our method draw on prior work, how to deploy these techniques together to achieve effective CoT length reduction without performance degradation is unclear and turns out to be non-trivial in our exploration. Thus, our main contributions lie in (1) providing the first systematic analysis of the optimization challenges that arise when applying length penalties to reduce COT length, (2) showing how to resolve these issues by carefully combining existing techniques into a principled and effective training recipe, and (3) demonstrating that the choice of optimization techniques is far more important than designing increasingly complex length rewards, (4) open the black box of why and how existing approaches improve training dynamics from the root.
>
> Specifically, prior studies on efficient reasoning focus on designing increasingly complex reward functions which they overlook the underlying optimization instability that arises when applying those length rewards. Our paper identifies this optimization bottleneck clearly and shows why naïvely applying length penalties with default optimization setups leads to degradation in performance.
>
> Building on this analysis, we curate and adapt techniques from DAPO and Hu et al. (2025), integrating them into DLER, a training recipe that achieves state of the art accuracy/length tradeoffs using the simplest possible length reward. Our work demonstrates that more complex designs of length penalties are not necessary: what matters is selecting the right optimization strategies and combining them coherently.

---

### Author Response · Authors · 2025-11-28
**A Friendly Reminder to Review the Responses We Provided**

Dear Reviewers,

I hope you’re doing well and having a wonderful Thanksgiving holiday. I just wanted to send a brief reminder that the rebuttal period will be ending soon. If you have any questions or would like clarification on any part of the work, please don’t hesitate to let us know! Thank you again for the time and effort you’ve put into reviewing our paper.

---

### Meta-Review · Area_Chair_i36K · 2026-01-10

**Summary:**

The primary reason is the lack of technical novelty. Reviewers pointed out that DLER’s core components (e.g., dynamic sampling, high clipping, batch normalization) are largely derived from prior works such as DAPO or Hu et al. (2025), making this work feel more like a strong engineering consolidation and reproduction report rather than a substantive methodological advance.

**Reviewer Concerns:**

Addressed: The authors demonstrated generalizability beyond math through coding task results  provided direct comparisons with the DAPO baseline showing DLER's superior accuracy-length tradeoff ; and clarified technical misunderstandings regarding truncation length and weight-merging

Outstanding: The fundamental concern remains methodological originality. Reviewers B7sj, nZtF, and wEUt maintained that despite the systematic analysis, the work essentially combines existing techniques and lacks the significant theoretical or methodological leap expected for ICLR

**Reviewer Scores:**

Reviewer B7sj: Likely to stay at 4.

Reviewer YCru: Likely to stay at 8.

Reviewer nZtF: Likely to stay at 4

Reviewer wEUt: Likely to stay at 4

---

### Decision · Program_Chairs · 2026-01-26

Reject